
# Stability analysis of geomagnetic baseline data obtained at Cheongyang observatory in Korea

Shakirah M. Amran[1,2,3], Wan-Seop Kim[1,2] , Heh Ree Cho[1]  and Po Gyu Park [1,2]

[1]Korea Research Institute of Standards and Science, 267 Gajeong-ro, Yuseong-gu, Daejeon 34113, Rep. of Korea
5  [2]University of Science and Technology, 217 Gajeong-ro, Yuseong-gu, Daejeon 34113, Rep. of Korea
[3]National Metrology Institute of Malaysia, Lot PT 4803 Bdr Baru Salak Tinggi, 43900 Sepang, Selangor, Malaysia

*Correspondence to*: Po Gyu Park (pgpark@kriss.re.kr)

10  **Abstract.** The stabillity of baselines produced by Cheongyang (CYG) observatory from a period of 2014 to 2016 is analysed. Step height of higher than 5 nT were found in *H* and *Z* components in 2014 and 2015 due to magnetic noise in the absolute measurement hut. In addition, a periodic modulation behaviour observed in the *H* and *Z* baseline curves was related to temperature variation of about 20 °C in the fluxgate magnetometer hut. The quality of the baselines was improved by correcting the discontinuity in the *H* and *Z* baselines. Moreover, the stability of the both baselines was also improved by about 15  6 nT/year from 10 nT/year by performing the temperature effect correction.

**Keywords.** Geomagnetic observatory, baseline, absolute measurement

## 1 Introduction

Geomagnetic observatories data are mainly used for monitoring of secular variations. Data with longer time series and and 20  greater absolute accuracy are very valuable for detailed monitoring of the secular variations. Relative and absolute measurements are conducted in order to obtain continuous and reliable geomagnetic data. Relative measurements recorded the variations of three independent components of geomagnetic field relative to baselines using a fluxgate magnetometer. Absolute measurements are conducted on a regular basis to measure magnetic direction, declination (*D*) and inclination (*I*). These absolute magnetic-direction data are then used to calibrate the fluxgate magnetometer, so as to compensate for its long term 25  drift and to calculate the calibration curve, called a baseline. The baseline values are derived from the difference between the absolute measurement results and the variation data provided by a fluxgate magnetometer.

Long-term behavior of baseline values provides an evidence for the stability of the fluxgate magnetometer operation. Good baseline stability makes monitoring of secular variations easier. Baseline values with frequent measurement points, small drift and low scatter indicate good quality data and a good performance of the observatory (McLean et al, 2004). Baseline variations 30  of 5 nT/year or less are recommended by INTERMAGNET (St-Louis, 2012).



In this study, we present the results of the stability analysis and improvement of the baseline data quality of the *H* and *Z* magnetic components by performing correction of temperature effect and an additional reduction of step height in the both baseline curves caused by artificial magnetic components in the absolute measurement.

## 2 Observatory Site and Instrumentation

The CYG observatory was built in a mountainous area about 5 km away from the main traffic road to reduce artificial magnetic noise.  The observatory contains five huts, separated more than 5 m form one another as shown in Fig. 1. In hut 1, a scalar magnetometer was installed for measurements of the total field intensity. A 3-axis fluxgate magnetometer was mounted on a marble pillar in hut 2 to measure magnetic field variations and its electronic unit was placed in hut 3. Hut 4 is containing a sturdy pillar for mounting of a fluxgate-theodolite used for absolute observations. The pillar served as the reference point for

the total field  intensity measurement. Hut 5 is used for control room where a computer controls data acquisition and transmission of the measured data via internet to a server of Korea Meteorological Administration (KMA).

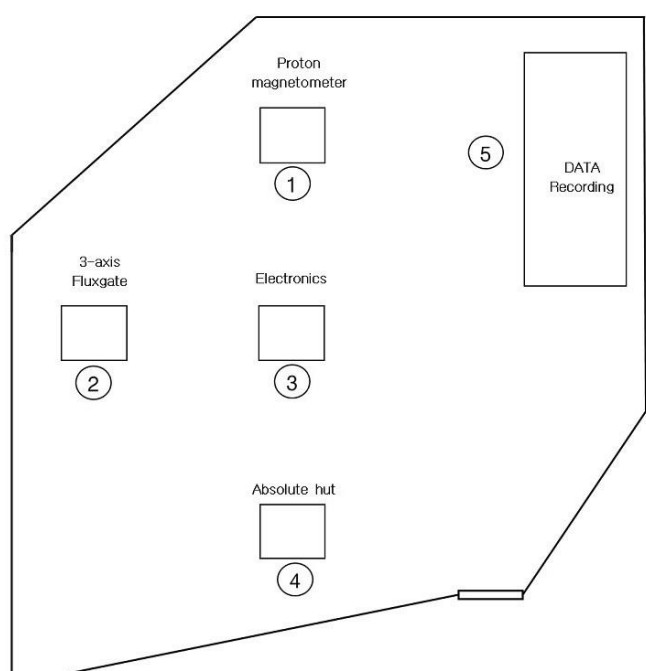

**Figure 1:**  Layout of the CYG observatory. A scalar magnetometer for total field intensity measurement was installed in hut

1. Hut 2 is used for installation of a 3-axis fluxgate magnetometer and its electronic was placed in hut 3. A pillar used for absolute measurement and served as reference point for total field intensity measurement was installed in hut 4. Hut 5 is used for control room for data acquisition and transmission to a server.



Geomagnetic variations in magnetic components *X*, *Y* and *Z* are recorded at a 1-sec sampling rate with a resolution of 0.1 nT by means of FGE 3- axis linear core fluxgate magnetometer from DTU Space Denmark. Sensors are located underground in a box to minimize temperature variations. The FGE electronics are placed in a separate hut to avoid magnetic interference. In addition to the fluxgate sensors, a scalar magnetometer (Overhauser-effect proton precession magnetometer (PPM)) model

GSM-19T from GEM Systems is independently installed for measurement of total field intensity (*F*). The total field values are recorded at every 5-sec with a 0.1 nT resolution.

The absolute measurements of *D* and *I* are conducted weekly using a non-magnetic theodolite (Zeiss 010A) with an integrated single-axis fluxgate (DTU model G). In each measurement session, four absolute measurements are performed on the basis of the magnetic field null-method. Total field intensity difference between the absolute measurement pillar and the

10 PPM pillar is measured using a Cs-He standard magnetometer with a 0.1 nT resolution. The site difference value is taken into account to correct the continous scalar readings with reference to the magnetic field value of the absolute measurement pillar.

To derive the baseline values, the absolute measurements are processed using the Java program GDASView developed by the British Geological Survey (BGS). Variation of the baseline values is fitted by piecewise polynomial up to 3$^{rd}$ order to minimise deviation of baselines (Clarke et.al., 2013).

## 15  3 Baseline variations

The observed baseline values of *D*, *H* and *Z* components from the measured *D*, *I* and *F* from 2014 to 2016 are shown in Fig. 2. Step heights of higher than 5 nT can be seen in *H* and *Z* baselines during 2014 and 2015. The first step with magnitude of approximately 5.2 nT was found in *H* on 1 July 2014. Another step with similar magnitude happened on 23 October 2014, pushing the *H* baseline further down and ends on 4 June 2015. On 12 June 2015, a jump of approximately 7.3 nT occurred,

brings the baseline to a new level and continues until end the of 2016. In addition, the *Z* baselines follow the same trend with approximately the same magnitude. The trend indicates that an offset was introduced to the baseline which caused a baseline shift from July 2014 until June 2015. Small step of approximately 0.3 arcmin in magntitude can be observed in *D* on 1 July 2014 but it does not show a similar trend with *H* and *Z*.





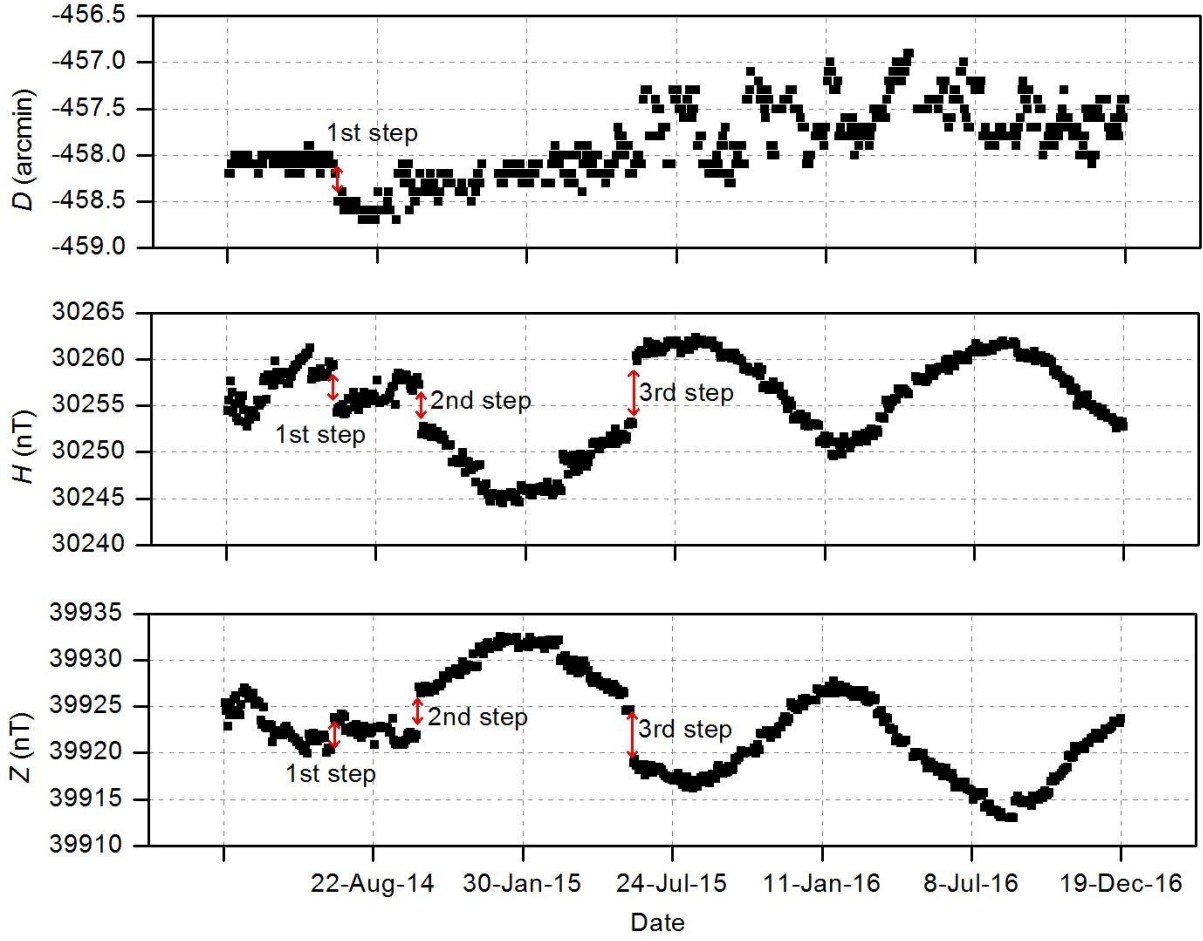

**Figure 2:** Observed *D*, *H* and *Z* baselines calculated from the measured *D*, *I* and *F* from 2014 to 2016. The first step in the *D*, *H* and *Z* baselines was occurred on 1 July 2014, second and third steps were found in *H* and *Z* on 23 October 2014 and 12 June 2015, respectively.

Observatory log book on June 2014 showed that LED light panels were installed in the absolute hut on top of the absolute measurement pillar close to the fluxgate sensor as shown in Fig. 3 and were later removed in June 2015. The period in which the LED panels were installed and removed is consistent with the period when the baseline shift occurred. The magnetic part from the LED panels caused the first jump in July 2014. The LED lights may not be immediately in use because of sufficient

10 light in the absolute hut during the summer months. However during October when the natural light was not sufficient, the LED lights and a battery pack generate magnetic field during the absolute measurements and cause an another step in the baselines in October. Upon removal of the LED panels in June 2015, the offset in the baselines was removed. Inspection on




the variations data does not indicate any steps during these period. Thus, it can be confirmed that the steps are caused by the error in the absolute measurements due to the installation of the magnetic LED light panels.

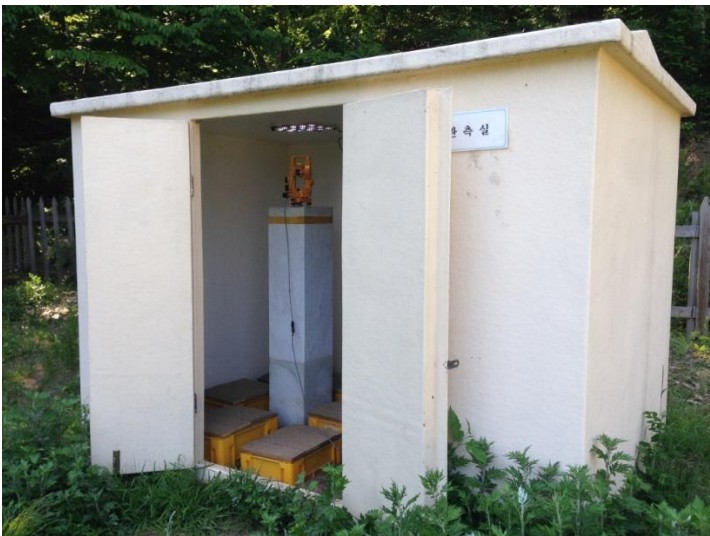

**Figure 3:** LED light panels installed in the absolute measurement hut.

Adjustments were made to the baselines in order to fix the steps. For the *D* baselines, the adjustment value is determined from the baseline difference immediately before and after the step on July 2014. The *D* baselines from 7 July 2014 to 4 June 2015 were then adjusted. For the *Z* component, the baseline difference immediately before and after each step was calculated.

The baseline difference $d_1$, $d_2$ and $d_3$ are calculated on July 2014, October 2014 and Jun 2015, respectively. The average of these differences $(d_1 + d_2)/2$ and $(d_2 + d_3)/2$ determine the size of the adjustments and was applied to the *H* and *Z* baselines from  7 July 2014 to 1 October 2014 and from 23 October 2014 to 4 June 2015, respectively.

We also checked for the stability of the fluxgate sensor mounting. Some sensors in the DTU single-axis fluxgate magnetometers are reported to give unstable readings of the offset due to loose ferromagnetic cores (Pederson and Matzka,

2012). The sensor instability can give a discrepancy in zero readings of absolute measurement. In order to check the loose core problem in the single-axis fluxgate used in the CYG, the sensor offset as well as the collimation angle from the *D* and *I* measurements were calculated. The sensor offset includeed the residual magnetism of the magnetometer and the offset of the electronics calculated from the D and the *I* circle readings according to Eq. (1) and (2):

$$s_{OD} = H \, sin[(East \; Down + East \; Up − West \; Down – West \; Up)/4], \qquad (1)$$

$$s_{OI} = F \, sin[(South \; Down + South \; Up − North \; Down – North \; Up)/4] , \qquad (2)$$

The collimation angle ε is the angle between the measurement axis of the magnetometer and the optical axis of the telescope in vertical plane. The angle is calculated from *D* and *I*  circle readings as below:



$$\varepsilon_D = (West\ Down + East\ Up - East\ Down - West\ Up \pm 360°)/(4 \cdot \tan I)\,, \qquad (3)$$

$$\varepsilon_I = (North\ Down + South\ Up - North\ Up - South\ Down)/4\,, \qquad (4)$$

The result in Fig. 4 shows that the sensor offset $S_{OD}$ and $S_{OI}$ agreed within 10 nT and the collimation angle $\varepsilon_D$ and $\varepsilon_I$ are constant within ± 1 arcmin . The sensor offset $S_{OD}$ and $S_{OI}$ as well as the collimation angle $\varepsilon_D$ and $\varepsilon_I$ do not show large discrepancies that can cause an error in the absolute measurement. It can be assumed that the sensors used in CYG are stable.

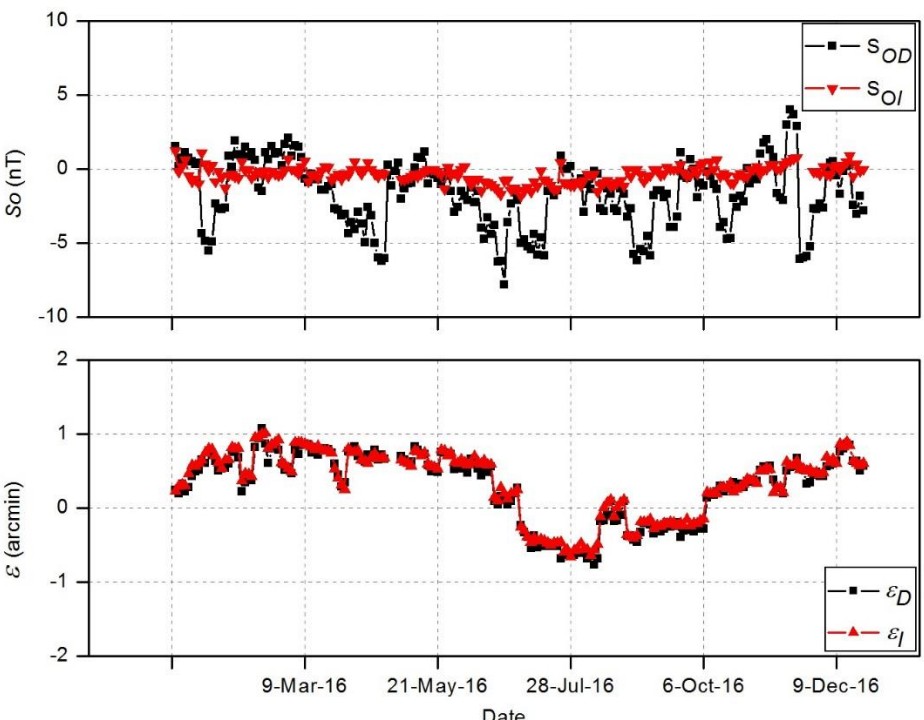

**Figure 4:** Sensor offset $S_O$ and collimation angle $\varepsilon$ calculated from the $D$ and $I$ measurements.

H and Z baselines after adjustments are presented in Fig. 5. The baseline shift in H and Z has reduced to within INTERMAGNET acceptable limit. H and Z baseline drifted within 10 nT in 2014 and 2015 and increased a few nT in 2016 due to secular variations. Dispersion of consecutive measurements is calculated and the values are well less than 1 nT in H and Z throughout the period. The standard deviation of the dispersion in both components is reduced to 30% in 2016. This shows that the quality of absolute measurements has improved over the period. However numbers of small jumps with magnitude of 2 nT are still noticeable in H and Z, probably caused by the error from the observer or handling of equipment during observation. As seen in Fig. 3, cable from the fluxgate sensor is hanging free and not fixed to the theodolite. The cable





and connector weight can pull down the sensor and caused a bit of movement during observations which can change the zero

readings of the fluxgate magnetometer.

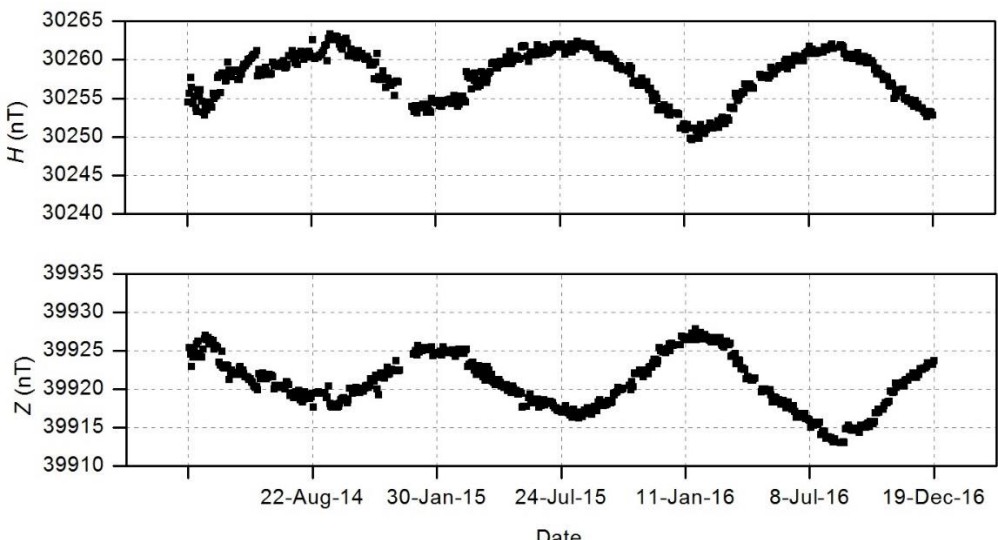

**Figure 5:** *H* and *Z* baselines after adjustment of steps.

Figure 6 presents the variations of absolute *D* measurement values. The absolute *D* decreased linearly with time with a rate of

4 arcmin/year. The accuracy of the measurement has also improved as seen from the scatter of the data. This is in contra with

the *D* baselines in which the values increased with time as seen in Fig. 2. The variations of D also increased approximately 0.7

arcmin in 2015 and 2016. We compare the absolute *D* values from the CYG and the *D* values calculated from the International

10   Geomagnetic Reference Field (IGRF) model. The results show a similar trend and the rate of change/year is approximately the

same. Thus, it can be assumed that the increase of *D* baseline and its variation values is not caused by the inaccurate absolute

measurement but rather from the drift of the fluxgate magnetometer.



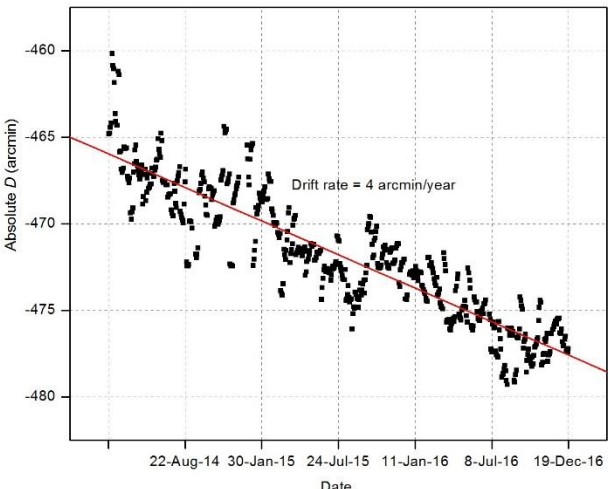

**Figure 6:** The variations of absolute *D* from 2014 to 2016.

Figure 7 shows the *D*, *H* and *Z* baselines plotted against temperature in fluxgate magnetometer sensor hut. *H* and *Z* baseline

5    show a yearly drift of more than 5 nT. Meanwhile *D* baselines show a yearly drift of approximately 1 arcmin/year. A large
temperature variation can be observed annualy, approximately 19 °C to 21 °C due to seasonal variation. This could not be
avoided since a temperature control system is not installed in the fluxgate magnetometer sensor hut. *H* and *Z* baselines show a
clear relation to temperature, implying that the drift of these baselines  are due to the large temperature changes in the fluxgate
magnetometer sensor hut. However, *D* baselines do not show a clear relation to temperature changes. The temperature

10    coefficients of the sensor part were estimated as 0.3 nT/°C for *H* and *Z*. These values are consistent with the values for fluxgate
sensor's specification. We could keep the *H* and *Z* baseline drift within 6 nT/year by performing the temperature correction as
shown in Fig. 8.

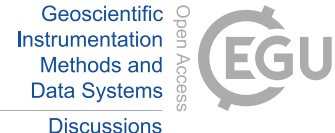



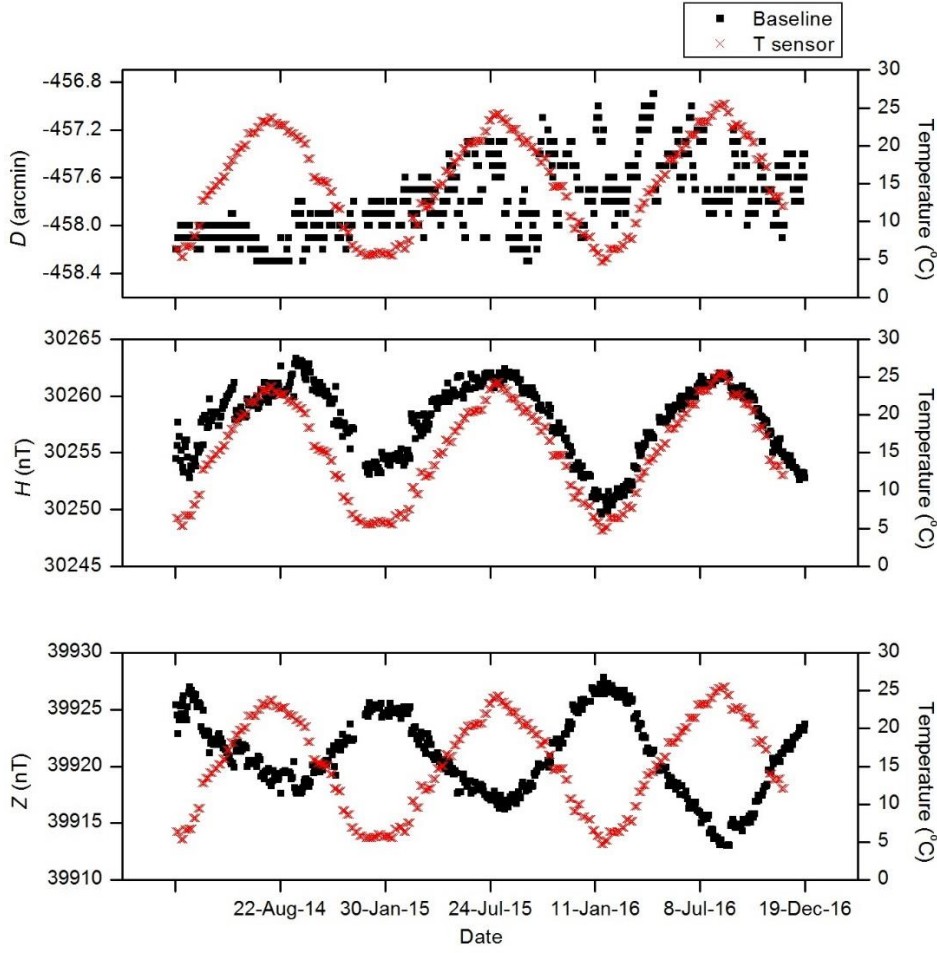

**Figure 7:** *D*, *H* and *Z* baselines plotted against temperature of fluxgate magnetometer sensor hut.



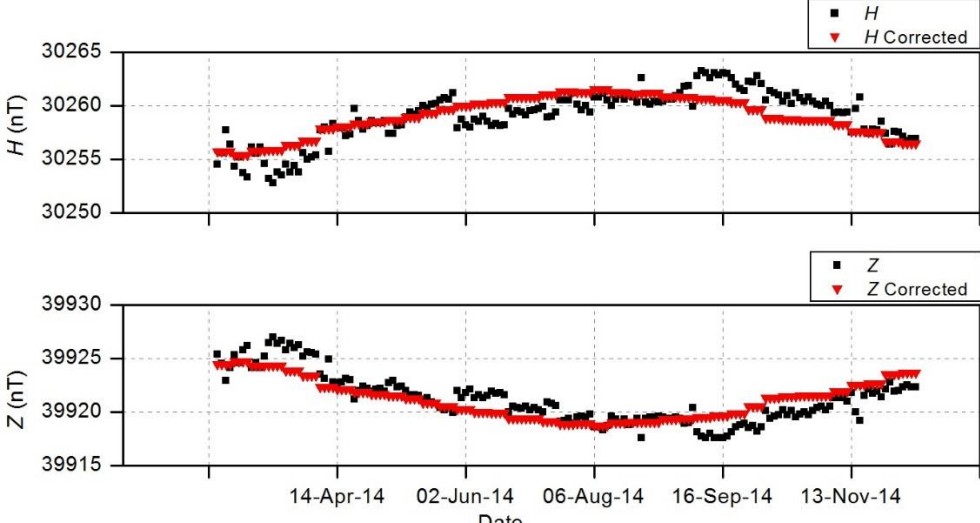

**Figure 8:** Results of temperature corrected in *H* and *Z* baselines.

## 4 Conslusion

Variations of baselines produced by the CYG observatory from a period of 2014 to 2016 are analysed. Steps of more than

5   5 nT were found in *H* and *Z* baselines causing a baseline shift form July 2014 to June 2015. The installation of the LED light

panels was identified as the reason for the jumps in the absolute measurement during this period. Steps are reduced to less than

5 nT after adjustments of the baselines.

The baselines produced by the CYG comply with the INTERMAGNET standards which shows the capability of CYG to

produce good quality data. .  The accuracy of the measurement has improved as seen from the scatter of the data. The baselines

drift can be further reduceed by performing the temperature correction. Better approach is to minimize temperature variations

in the sensor and electronics using a temperature control system as adopted by most observatories. Factors such as stability of

fluxgate-theodolite, levelling and magnetic cleanliness can affect the accuracy of the absolute measurement should always be

checked during observations to avoid unnecessary steps from occurring in the absolute measurement.

**Acknowledgement**

This work was supported by Korea Meteorological Administration. The author would like to thank Ms Orsi Baillie from British

Geological Survey for her continuous support and assistance on data processing and Mr Tero Raita from Sodankylä

Geophysical Observatory for checking of data.





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
