# Peer review of "Stability analysis of geomagnetic baseline data obtained at Cheongyang observatory in Korea"

_Geoscientific Instrumentation, Methods and Data Systems, 2017_

## Referee Comment (RC1) · Anonymous Referee #1 · 20 Mar 2017

GENERAL COMMENTS.

This paper can be interesting especially for people, who are conducting routine observations of geomagnetic field. The problems described in that paper are more common, especially in new-established observatories. The paper draws especially attention on problems related to observations of slow changes of the earth's magnetic field i.e. geomagnetic secular variations. We know that accuracy of observations of secular variations is mainly determined by quality of baseline determining. The authors of the article rightly note, that special attention should be paid to an unexpected jumps on baselines. Such jumps were observed on 2014 and 2015 baselines of Cheongyang observatory. A cause can be both on the side of recording of magnetic variations and

absolute control. The analysis carried out in the paper shows that the reason was in the absolute house. The paper presents the method of baseline correction. However there is no solid evidence, that correction is appropriate for all elements (H,D,Z). Comparison D from CYG and D from IGRF is only partial evidence It is not understandable that no analog information is given on H and Z elements although they have greater baseline jumps. I think a good evidence would be a comparison with neighboring observatories (e.g. KAK, KNY, MMB).

SPECIFIC COMMENTS.

Most items listed in the section "References" are not referred in the text of the paper. That is completely unacceptable.

Page 1, line 28 The word "easier" should be rather replaced with "more accurate" or similar.

Page 4, line 9 The word "immediately" should be rather replaced with other word e.g. "permanently".

Page 2, line 10 Now is "is used for control room". Should be rather "is used as control room".

Page 3, lines 2&3, lines 22@23 In my opinion there is contradiction between the following sentences: "The first step with magnitude of approximately 5.2 nT was found in H on 1 July 2014." (lines 2&3) and "Small step of approximately 0.3 arcmin in magntitude can be observed in D on 1 July 2014 but it does not show a similar trend with H and Z." (lines 22@23)

TYPING ERRORS and TECHNICAL CORRECTIONS

Page 2, line 7 Now is "form". Should be "from"

Page 3, line 22 magnitude instead of magntitude

---

## Referee Comment (RC2) · Anonymous Referee #2 · 29 Mar 2017

The paper discusses the instabilities observed in baselines of Cheongyang observatory for a period of two years. The authors have analyzed the cause for the shifts and the cyclical pattern observed in the baseline. The shifts were found to be caused due to the installation of LED lamp in the Absolute room. The authors ascribe the cyclical pattern observed in the baseline to the temperature affect on the fluxgate variometer and the problem was corrected to a large extent by introducing a temperature correction. The problem of baseline instability discussed in the paper is encountered in many observatories and is of interest to the magnetic observatory community.

The paper can be further improved by giving more explanation on the following points and suggestions.

[Figure]

1) A scatter plot of temperature versus baseline for each component can be included in the paper.

2) In Figure 7, for Z component for the year 2016, the temperature varies from 5°C to 25°C approximately whereas the Z component base line for the corresponding period varies from 39913 to 39927 nT approximately. This gives a temperature coefficient of 0.7 nT/ °C.

Similarly for H component in the same Figure 7 during the period Jul 2015 to Dec 2016, when the temperature range was 20°C, the corresponding H baseline varied from 30250 nT to 30262 nT. This gives a temperature coefficient of 0.6 nT/ °C.

As per the FGE fluxgate manual, the temperature coefficient of sensor is less than 0.3 nT/°C. The value of 0.7 nT/ °C (for Z) and 0.6 nT/ °C (for H) observed Cheongyang observatory is far higher than the temp. coefficients specifications normally observed. This along with a complete absence of temperature sensitivity on the D Sensor is intriguing.

The baseline of H and Z appears to become more sensitive to temperature over time as seen from the same Figure 7. The authors may explain how they took care of this issue of varying sensitivity in the temperature correction applied and also explain how they arrived at a temp. coefficient estimate of 0.3 nT / °C .

3) Please check whether the temperature effect observed in the baseline is due to temperature affecting the absolute instruments in absolute room.

1) Please ensure that all entries in the bibliography are mentioned in the text.

2) Typo errors may be corrected.

Line no. 19 page 1 and and Line no. 11 page 3 Continous Line no. 22 page 3 magntitude Line no. 17 page 5 includeed Line no. 6 page 8 annualy Line no. 10 page 10 reduceed

---

## Author Comment (AC1) · 6 May 2017

The authors thank to the Peer Reviewers for his/her valuable comments to the manuscript. All changes done in the light of the reviewer's comment are highlighted in the revised version.

GENERAL COMMENTS

1. The paper presents the method of baseline correction. However there is no solid evidence, that correction is appropriate for all elements (H,D,Z). Comparison D from CYG and D from IGRF is only partial evidence It is not understandable that no analog information is given on H and Z elements although they have greater baseline jumps.

[Figure]

I think a good evidence would be a comparison with neighboring observatories (e.g. KAK, KNY, MMB).

Author's response: The authors have made a comparison with KAK data. The KAK data show that there are no similar jumps in all baseline components. Fig. 4 was added to show the KAK baselines for comparison. A brief explanation to the figure was added too.

SPECIFIC COMMENT

2. Most items listed in the section "References" are not referred in the text of the paper. That is completely unacceptable.

Author's response: All the references are cited in the text.

3. Page 1, line 28 The word "easier" should be rather replaced with "more accurate" or similar.

Author's response: Corrected as reviewer suggested.

4. Page 4, line 9 The word "immediately" should be rather replaced with other word e.g. "permanently".

Author's response: Corrected as reviewer suggested.

5. Page 2, line 10 Now is "is used for control room". Should be rather "is used as a control room".

Author's response: Corrected as reviewer suggested.

6. Page 3, lines 2&3, lines 22@23 In my opinion there is contradiction between the following sentences: "The first step with magnitude of approximately 5.2 nT was found in H on 1 July 2014." (lines 2&3) and "Small step of approximately 0.3 arcmin in magnitude can be observed in D on 1 July 2014 but it does not show a similar trend with H and Z." (lines 22@23) .
Author's response: We wanted to explain that the first step can be found in DHZ components on 1 July 2014. But after July 2014, no more steps can be seen in D baselines compared to the H and Z baselines, in which two more steps happened on 23 Oct 2014 and 4 Jun 2015. We have rephrased the sentences to provide better explanation. We have also made a correction on the date of first step which is 7 July 2014 and not 1 July 2014.

TYPING ERRORS and TECHNICAL CORRECTIONS

7. Page 2, line 7 Now is "form". Should be "from"

Author's response: Corrected as reviewer suggested.

8. Page 3, line 22 magnitude instead of magntitude

Author's response: Corrected as reviewer suggested.

Please also note the supplement to this comment:
http://www.geosci-instrum-method-data-syst-discuss.net/gi-2017-8/gi-2017-8-AC1-
supplement.pdf
* * *
[Figure]

**Supplement:**

**Stability analysis of geomagnetic baseline data obtained at Cheongyang observatory in Korea**

Shakirah M. Amran[1,2,3], Wan-Seop Kim[1,2] , Heh Ree Cho[1]  and Po Gyu Park [1,2]

[1]Korea Research Institute of Standards and Science, 267 Gajeong-ro, Yuseong-gu, Daejeon 34113, Rep. of Korea
5 [2]University of Science and Technology, 217 Gajeong-ro, Yuseong-gu, Daejeon 34113, Rep. of Korea
[3]National Metrology Institute of Malaysia, Lot PT 4803 Bdr Baru Salak Tinggi, 43900 Sepang, Selangor, Malaysia

*Correspondence to*: Po Gyu Park (pgpark@kriss.re.kr)

10  **Abstract.** The stability of baselines produced by Cheongyang (CYG) observatory from a period of 2014 to 2016 is analysed. Step heights of higher than 5 nT were found in $H$ and $Z$ components in 2014 and 2015 due to magnetic noise in the absolute measurement hut. In addition, a periodic modulation behaviour observed in the $H$ and $Z$ baseline curves was related to annual temperature variation of about 20°C in the fluxgate magnetometer hut. Improvement in data quality was evidenced by a small dispersion between successive measurements from Jun 2015 to end of 2016. Moreover, the baseline was also improved by
15  correcting the discontinuity in the $H$ and $Z$ baselines.

**Keywords.** Geomagnetic observatory, baseline, absolute measurement

**1 Introduction**

Geomagnetic observatories data are mainly used for monitoring of secular variations (Jankowski and Sucksdorff, 1996).  Data
20  with longer time series and  greater absolute accuracy are very valuable for detailed monitoring of the secular variations. In order to obtain continuous and reliable geomagnetic data, relative and absolute measurements are conducted under carefully controlled conditions. Relative measurements recorded the variations of three independent components of geomagnetic field relative to baselines using a fluxgate magnetometer. Absolute measurements are conducted on a regular basis to measure magnetic direction, declination ($D$) and inclination ($I$). The baseline values are derived from the difference between the absolute
25  measurement results and the variation data provided by a fluxgate magnetometer.

Baseline values with frequent measurement points, small drift and low scatter indicate good quality data and a good performance of the observatory (McLean et al., 2004). In addition, good baseline stability makes monitoring of secular variations more accurate (Reda et.al, 2011). Baseline variations recommended by INTERMAGNET for the participating observatories are of 5 nT/year or less (St-Louis, 2012). In practice, error factors affecting the absolute measurement instrument,
30  the magnetometer such as temperature, pier tilts, ageing of electronics components, etc. and the observational procedure can caused a large drift in baseline. In this study, we present the results of the stability analysis on the observed baselines obtained from the period of 2014 to 2016. Above all, the baseline data quality of the $H$ and $Z$ components was improved by correcting

the step height in the baseline curves caused by artificial magnetic components in the absolute measurements. In addition, we also analysed the temperature effect observed in the baseline as well as the quality of the absolute measurements obtained at Cheongyang observatory.

**2 Observatory Site and Instrumentation**

Cheongyang geomagnetic observatory (IAGA code CYG, latitude 36.370°N, longitude 126.854°E, elevation 165 meters), South Korea has been in operation since 2009 and gained official INTERMAGNET magnetic observatory (IMO) status in December 2013. The CYG observatory was built in a mountainous area about 5 km away from the main traffic road to reduce artificial magnetic noise as shown in Fig. 1. The observatory contains five huts, separated more than 5 m from one another. In hut 1, a scalar magnetometer was installed for measurements of the total field intensity. A 3-axis fluxgate magnetometer was mounted on a marble pillar in hut 2 to measure magnetic field variations and its electronic unit was placed in hut 3. Hut 4 is containing a sturdy pillar for mounting of a fluxgate-theodolite used for absolute observations. The pillar served as the reference point for the total field intensity measurement. Hut 5 is used as a control room where a computer controls data acquisition and transmission of the measured data via internet to a server of Korea Meteorological Administration (KMA).

[Figure]

**Figure 1:** Location and site layout of the CYG observatory. The upper left panel shows the mountainous area where CYG observatory is located and lower panel shows the observatory site. The right panel shows the layout of the observatory site. A scalar magnetometer for total field intensity measurement was installed in hut 1. Hut 2 is used for installation of a 3-axis fluxgate magnetometer, and its electronics was

[revised manuscript text omitted]

To verify a consistence of the steps, the variation data was checked. But there are no observable steps in the variation data. In addition, we compared the CYG baselines with Kakioka (KAK) observatory data for the same period as shown in Fig. 4. Although steps are noticeable in KAK baselines on October 2014, the magnitude is small approximately 1 nT. Furthermore, no large steps can be found in KAK baselines on July 2014 and June 2015. Thus, it can be confirmed that steps happened in CYG are due to the artificial noise which caused an error in the absolute measurements.

[Figure]

**Figure 4**: KAK baselines values obtained from 2014 to 2016. The data of the KAK was provided by the Kakioka Magnetic Observatory, Japan Meteorological Agency.

5     Adjustments were made to the baselines in order to fix the steps. For the $D$ baselines, the adjustment value is determined from the baseline difference immediately before and after the step on July 2014. The $D$ baselines from 7 July 2014 to 4 June 2015 were then adjusted. For the $H$ and $Z$ component, the baseline difference $d_1$ were calculated on July 2014 and was applied to the $H$ and $Z$ baselines from 7 July 2014 to 1 October 2014. Then, the next baselines difference $d_2$ and $d_3$ are calculated on October 2014 and Jun 2015, respectively and the average $(d_2 + d_3)/2$ was applied from 23 October 2014 to 4 June 2015. $D$, $H$
10  and $Z$ baselines after adjustments are presented in Fig. 5. The baseline shift in $H$ and $Z$ has reduced to approximately 2 nT, while $D$ baselines show a noticeable improvement after the correction.

[Figure]

**Figure 5:** *D*, *H* and *Z* baselines before adjustment (solid rectangular) and after adjustment (cross) of steps.

We also checked for the stability of the fluxgate sensor mounting. Some sensors in the DTU single-axis fluxgate
magnetometers are reported to give unstable readings of the offset due to loose ferromagnetic cores (Pederson and Matzka,
2012). The sensor instability can give a discrepancy in zero readings of absolute measurement. In order to check the loose core
problem in the single-axis fluxgate used in the CYG, the sensor offset as well as the collimation angle from the *D* and *I*
measurements were calculated. The sensor offset included the residual magnetism of the magnetometer and the offset of the
electronics calculated from the D and the *I* circle readings according to Eq. (1) and (2) (Turbitt, 2004):

$$s_{OD} = H \sin[(East\ Down + East\ Up - West\ Down - West\ Up)/4], \tag{1}$$

$$s_{OI} = F \sin[(South\ Down + South\ Up - North\ Down - North\ Up)/4], \tag{2}$$

The collimation angle $\varepsilon$ is the angle between the measurement axis of the magnetometer and the optical axis of the telescope in vertical plane. The angle is calculated from $D$, and $I$ circle readings as below:

$$\varepsilon_D = (West\ Down + East\ Up - East\ Down - West\ Up \pm 360°)/(4 \cdot \tan I) , \tag{3}$$

$$\varepsilon_I = (North\ Down + South\ Up - North\ Up - South\ Down)/4 , \tag{4}$$

The result in Fig. 6 shows that the sensor offset $S_{OD}$ and $S_{OI}$ agreed within 10 nT and the collimation angles $\varepsilon_D$ and $\varepsilon_I$ are constant within ± 1 arcmin. The sensor offset $S_{OD}$ and $S_{OI}$ as well as the collimation angle $\varepsilon_D$ and $\varepsilon_I$ do not show large discrepancies that can cause an error in the absolute measurement. Thus, the analysis above implies that the sensors used in CYG are stable.

[Figure]

**Figure 6:** (top panel) Sensor offset $S_{OD}$ for the declination and $S_{OI}$ for the inclination and (bottom panel) collimation angle $\varepsilon_D$ for the declination and $\varepsilon_I$ for the inclination.

**4 Baseline variations**

Figure 7 presents the $D$, $H$ and $Z$ baselines and daily mean temperature in the fluxgate magnetometer sensor and electronics hut, respectively. The temperature effect on the fluxgate magnetometer measurement can be clearly seen on $H$ and $Z$ baselines

although *D* component does not show a clear relation to temperature changes. The fluxgate sensor and electronics experienced significant temperature swings, as much as 20°C annually. The daily temperature variations between 0. 2°C to 3°C were observed in the sensor hut and 0.6°C to 5°C in the electronics hut. In order to check the temperature effect on the absolute instruments, the declination and inclination values as a function of temperature obtained in the absolute hut during the observation are plotted in Fig. 8. Both *D* and *I* show a small change with the temperature rate of -0.001°/°C. We could assume that the temperature effect observed in baselines is mainly due to the fluxgate magnetometer.

[Figure]

**Figure 7**: *D*, *H* and *Z* baselines and (lowest panel) temperature of the fluxgate magnetometer sensor and the electronic huts.

[Figure]

**Figure 8**: (top panel) Declination and (bottom panel) inclination value plotted against temperature in the absolute hut.

Figure 9 presents the temperature dependent variations of the *D*, *H* and *Z* baselines depicted as a function of temperature in the sensor hut from 2014 to 2016 and their calculated temperature coefficients. The baselines show an increasing amplitude with temperature indicating that the fluxgate has large temperature coefficient mainly on *H* and *Z* components. Study by Csontos et.al (2007) proved that most of the fluxgate magnetometers have large temperature coefficient and their behaviour depends significantly on the amplitude of temperature change. The temperature coefficients of the *H* baseline increases from 0.3 nT/°C in 2014 to 0. 6 nT/°C in 2016. Whereas the temperature coefficients of the *Z* baseline varies from -0.3 nT/°C in 2014 to -0.7 nT/°C in 2016. Temperature influence on *D* baselines is considerably low with respect to those of the *H* and *Z* baselines and changes from -0.008 arcmin/°C in 2014 to 0.002 arcmin/°C in 2016. These varying sensitivity in the temperature coefficient limits a possibility to determine a general correction factor for temperature effect. Hence, the use of a temperature stabilized environment is the best way to achieve very accurate measurement (Csontos, et. al, 2007).

[Figure]

Figure 9: Temperature coefficient of *D*, *H* and *Z* baselines for (a) 2014, (b) 2015, (c) 2016.

The corrected baselines in Fig. 7 show a better stability with time. The dispersion of consecutive measurements is well less than 1 nT in *H* and *Z* throughout the period, with standard deviation reduced by 30% in 2016 indicating that the quality of the absolute measurement has improved over the period. Although *D* baselines show a larger deviation in 2015 and 2016, the accuracy of absolute *D* measurement has improved as seen from the scatter of the data shown in Fig. 10. The standard deviation of dispersion has reduced by 20% in 2015 and 2016. The absolute *D* also show a decreasing value with time, in contrary with *D* baselines. Comparison of absolute the *D* values with the International Geomagnetic Reference Field (IGRF) model shows a similar trend and rate of change which is approximately -4 arcmin/year (dashed line in Fig. 10).

[Figure]

Figure 10: Comparison of *D* baselines (circle with cross) and absolute *D* (triangle) from 2014 to 2016. The dashed line is the linear fit to the absolute measurement.

**4 Conslusion**

Variations of baselines produced by the CYG observatory from a period of 2014 to 2016 are analysed. Steps of more than 5 nT were found in *H* and *Z* baselines causing a baseline shift from July 2014 to June 2015. The installation of the LED light panels was identified as the reason for the jumps in the absolute measurement during this period. Steps are reduced to less than 5 nT after adjustments of the baselines. Generally, the baselines produced by the CYG comply with the INTERMAGNET standards which shows the capability of CYG to produce good quality data. The quality of the absolute measurement has improved with time as seen by the scatter of the data.

Temperature variation, ageing of electronic components, pier tilts, etc. are the known factors that can affect the long term stability of baselines. The temperature effect was supposed to be a major reason for the large drift in the CYG baselines. Use of temperature stabilized environment is the best way to minimize the temperature effect of the fluxgate magnetometer and to achieve accurate measurements. Observational procedure such as levelling, target readings, stability of fluxgate-theodolite, magnetic cleanliness and etc. can affect the accuracy of the absolute measurement and should always be checked during observations to avoid unnecessary steps from occurring in the absolute measurement.

**Acknowledgement**

We would like to thank Ms. Orsi Baillie from British Geological Survey for her continuous support and assistance on data processing. Special thanks to Mr. Tero Raita from Sodankylä Geophysical Observatory for his time and helpful advice in checking the baselines. We also like to thank Kakioka Magnetic Observatory, Japan Meteorological Agency for providing the KAK baseline data.

This work was funded by the Korea Meteorological Administration Research and Development Program under Grant KMIPA2015-3030.

---

## Author Comment (AC2) · 6 May 2017

The authors thank to the Peer Reviewers for his/her valuable comments to the manuscript. All changes done in the light of the reviewer's comment are highlighted in the revised version.

1. A scatter plot of temperature versus baseline for each component can be included in the paper.

Author's response: The scatter plot of temperature versus baseline for each component has been included and denoted as Fig. 8.

2. In Figure 7, for Z component for the year 2016, the temperature varies from 5 degC

to 25 degC approximately whereas the Z component base line for the corresponding period varies from 39913 to 39927 nT approximately. This gives a temperature coefficient of 0.7 nT/degC. Similarly for H component in the same Figure 7 during the period Jul 2015 to Dec 2016, when the temperature range was 20 degC, the corresponding H baseline varied from 30250 nT to 30262 nT. This gives a temperature coefficient of 0.6 nT/ degC. As per the FGE fluxgate manual, the temperature coefficient of sensor is less than 0.3 nT/degC. The value of 0.7 nT/degC (for Z) and 0.6 nT/degC (for H) observed Cheongyang observatory is far higher than the temp. coefficients specifications normally observed. This along with a complete absence of temperature sensitivity on the D Sensor is intriguing. The baseline of H and Z appears to become more sensitive to temperature over time as seen from the same Figure 7. The authors may explain how they took care of this issue of varying sensitivity in the temperature correction applied and also explain how they arrived at a temp. coefficient estimate of 0.3 nT /degC .

Author's response: We are suggesting that by doing the temperature correction, the baselines variation can be minimized. We used the 2014 data to estimate the temperature correction for the H and Z baselines which give us about 0.3 nT/degC. However, 2015 and 2016 baselines shows that our fluxgate magnetometer indeed exhibits a large temperature coefficient, more than the normal specification of the FGE fluxgate. And the value increased with increasing amplitude of temperature change. Therefore, it is difficult to determine a general temperature coefficient as a correction factor. We explained this in the revised manuscript. And we also rework our abstract to reflect this. Currently, we did not performed the temperature correction in our baselines but we plan to install the temperature control system in the in the near future to minimize this temperature effect.

3. Please check whether the temperature effect observed in the baseline is due to temperature affecting the absolute instruments in absolute room.

Author's response: We have plotted DI against temperature in the absolute room shown

in Fig. 8 of the revised manuscript. The D and I show only a small change with the temperature in the absolute room. Therefore we assume that the temperature effect observed in baselines is not due to the temperature affecting the absolute instruments in the absolute room. We briefly explained this in the revised manuscript.

4. Please ensure that all entries in the bibliography are mentioned in the text.

Author's response: The bibliography was cited in the text.

5. Typo errors may be corrected. Line no. 19 page 1 and and, Line no. 11 page 3 Continous, Line no. 22 page 3 magntitude, Line no. 17 page 5 includeed, Line no. 6 page 8 annualy Line no. 10 page 10 reduceed

Author's response: Corrected as reviewer suggested.

Please also note the supplement to this comment:
http://www.geosci-instrum-method-data-syst-discuss.net/gi-2017-8/gi-2017-8-AC2-supplement.pdf